# Effects of Two *Bacillus Velezensis* Microbial Inoculants on the Growth and Rhizosphere Soil Environment of *Prunus davidiana*

**DOI:** 10.3390/ijms232113639

**Published:** 2022-11-07

**Authors:** Huimin Shi, Lanxiang Lu, Jianren Ye, Lina Shi

**Affiliations:** Co-Innovation Center for Sustainable Forestry in Southern China, College of Forestry, Nanjing Forestry University, Nanjing 210037, China

**Keywords:** *Bacillus velezensis*, bacterial inoculant, growth-promoting function, rhizosphere, soil bacterial community

## Abstract

Microbial inoculants, as harmless, efficient, and environmentally friendly plant growth promoters and soil conditioners, are attracting increasing attention. In this study, the effects of *Bacillus velezensis* YH-18 and *B. velezensis* YH-20 on *Prunus davidiana* growth and rhizosphere soil bacterial community in continuously cropped soil were investigated by inoculation tests. The results showed that in a pot seedling experiment, inoculation with YH-18 and YH-20 resulted in a certain degree of increase in diameter growth, plant height, and leaf area at different time periods of 180 days compared with the control. Moreover, after 30 and 90 days of inoculation, the available nutrients in the soil were effectively improved, which protected the continuously cropped soil from acidification. In addition, high-throughput sequencing showed that inoculation with microbial inoculants effectively slowed the decrease in soil microbial richness and diversity over a one-month period. At the phylum level, Proteobacteria and Bacteroidetes were significantly enriched on the 30th day. At the genus level, *Sphingomonas* and *Pseudomonas* were significantly enriched at 15 and 30 days, respectively. These bacterial phyla and genera can effectively improve the soil nutrient utilization rate, antagonize plant pathogenic bacteria, and benefit the growth of plants. Furthermore, inoculation with YH-18 and inoculation with YH-20 resulted in similar changes in the rhizosphere microbiome. This study provides a basis for the short-term effect of microbial inoculants on the *P. davidiana* rhizosphere microbiome and has application value for promoting the cultivation and production of high-quality fruit trees.

## 1. Introduction

*Prunus davidiana* is a small tree or shrub belonging to the subfamily Prunoideae of the family Rosaceae [1]. It is commonly used as a grafting rootstock for peaches and plums in northern China because of its well-developed root system, its strong resistance to cold, drought, and saline alkalinity, and its adaptability to the environment [2,3]. Intensive production plays an important role in agricultural production in China, and most of the peach seedlings now being cultivated in intensive production are grafted [4]. However, intensive cultivation, years of continuous cropping, and the excessive use of chemical fertilizers often occur in this system [5]. Long-term irrational cultivation may lead to soil deterioration, the accumulation of self-toxic substances, and an imbalance of microbial communities, resulting in nutrient imbalance and the degradation of the soil, thus affecting crop yield and quality [6,7,8].

Soil is a complex and dynamic environment in which microbial communities control material circulation and energy flow, provide many ecosystem services for soil, and are considered important regulators of plant and agricultural ecological environments [9]. Soil microorganisms are widely distributed among plant roots, and their integration into the host plant’s beneficial bacterial community contributes to value-added soil nutrient cycling and high nutrient use efficiency [10]. Many studies have shown that microbial inoculants can induce changes in the plant rhizosphere microbial community and improve soil fertility, thus improving the soil environment and crop growth and reducing the pollution caused by unreasonable cultivation [11,12]. Xiao et al. [13] indicated that an inoculant could improve rice yield by mediating rare bacteria in the microbial community after co-inoculation with *Rhodopseudomonas palustris* and *B. subtilis*. Chen et al. [14] showed that a *B. subtilis* inoculant changed the relative abundance of the dominant soil phylum, increased the available nitrogen in the soil, and increased the wheat yield by 33.4%. Wang et al. [15] found that inoculation treatments with BIO-1 and BIO-2 significantly enriched beneficial microorganisms such as *Sphingomonas*, *Bacillus*, *Nocardioides*, *Rhizobium*, *Streptomyces*, *Pseudomonas* and *Microbacterium* and the therapeutic effects on wheat diseases were as high as 82.5% and 83.9%, respectively. Thus, the application of microbial inoculants has been considered an effective strategy to overcome the obstacles of crop succession, improve the structure of microbial communities, and maintain their beneficial functions [16,17].

*B. velezensis* is a widely reported plant growth-promoting rhizobacteria (PGPR) and a beneficial endophyte. Inoculation with *B. velezensis* can promote the growth of plants [18] and prevent many plant diseases [19,20,21,22]. Furthermore, *B. velezensis* can form stable endospores that help them survive in the preparation of bacterial bioinoculants [23]. Therefore, the application of *B. velezensis* in agricultural systems is becoming increasingly extensive [24]. Through previous experiments, the Forest Pathology Laboratory, Nanjing Forestry University, found that both YH-18 and YH-20 had a good ability to dissolve inorganic phosphorus. YH-18 had good nitrogen fixation ability and salt tolerance, YH-20 produced a high amount of indole-3-acetic acid (IAA), and both strains could colonize the plant rhizosphere [25,26]. In addition, the antibacterial proteins, lipopeptides, and volatile substances produced by YH-18 and YH-20 were found to effectively hinder the normal growth of plant pathogenic bacteria and to promote growth and disease resistance in many plants. These two plant growth-promoting (PGP) strains demonstrated good performance [26,27].

*B. velezensis*, as a PGPR bacterium, has good growth-promoting and disease-resistant abilities, but its influence on *P. davidiana* growth and rhizosphere soil microecology has not been reported. This study has three objectives: (1) to investigate the effects of inoculation with two *B. velezensis* inoculants on the growth of *P. davidiana* plants; (2) to determine the influence of two *B. velezensis* inoculants on changes in soil nutrient contents; and (3) to explore the regulatory effects of different inoculants on the diversity of the *P. davidiana* rhizosphere soil bacterial community and the key phenotypes of the microbial community.

## 2. Results

### 2.1. Effect of Inoculation on Peach Tree Growth

Inoculation with microbial inoculants influenced the growth characteristics of *P. davidiana* seedlings (Figure 1). Thirty days after inoculation, the YH-18 microbial inoculant significantly increased the ground diameter net growth by 93.1% compared to the CK group (*p* < 0.05). Furthermore, the chlorophyll content and leaf area increased by 7.90% and 10.27%, respectively. The YH-20 microbial inoculant significantly increased the ground diameter net growth by 120.68% (*p* < 0.05) and increased the chlorophyll content and leaf area by 7.11% and 8.79%, respectively.

Ninety days after inoculation, the YH-18 microbial inoculant significantly increased the ground diameter net growth, height net growth, and leaf area by 12.90%, 5.69%, and 20.41%, respectively, compared to the CK group (*p* < 0.05). The chlorophyll content increased by 2.79%. The YH-20 microbial inoculant significantly increased the ground diameter net growth by 10.39% (*p* < 0.05) and increased the chlorophyll content and leaf area by 3.85% and 19.02%, respectively.

One hundred and eighty days after inoculation, the YH-18 microbial inoculant significantly increased the ground diameter net growth and height net growth by 22.10% and 7.64%, respectively, compared to the CK group (*p* < 0.05). The YH-20 microbial inoculant significantly increased the height net growth by 4.59% (*p* < 0.05) and increased the ground diameter net growth by 4.52%.

### 2.2. Effect of Inoculated Microbial Agents on the Soil Nutrient Content

Thirty days after inoculation, the YH-18 microbial inoculant increased the available nitrogen (AN) in the rhizosphere soil by 13.78% compared to the CK group. The YH-20 microbial inoculant significantly increased AN and available potassium (AK) by 29.7% and 28.05%, respectively (*p* < 0.05). The pH value of the soil treated with YH-18 and YH-20 significantly decreased by 0.14 and 0.17, respectively, compared with the CK group (*p* < 0.05). Ninety days after inoculation, the YH-18 microbial inoculant increased available phosphorus (AP) in the rhizosphere by 11.15% compared to the CK group. The YH-20 microbial inoculant significantly increased AN and AP by 22.42% and 36.28%, respectively (*p* < 0.05). The pH value of the soil treated with YH-18 and YH-20 significantly decreased by 0.2 and 0.22, respectively (*p* < 0.05). Within 90 days of inoculation, AN, organic matter (OM), and soil pH decreased gradually; AK first increased and then decreased significantly, while AP first decreased and then increased significantly in the control group (Figure 2).

### 2.3. Correlation Analysis of Seedling Growth and Soil Environmental Parameters

Thirty days after inoculation, Pearson’s correlation analysis of the seedling growth correlation indicators and the soil nutrient content showed that OM and soil pH were significantly positively correlated with plant height growth. Ground diameter growth was significantly positively correlated with AK. Soil pH was significantly positively correlated with leaf area (Table 1).

### 2.4. Study of a Diversity and Its Correlation with Soil Properties after Inoculation

Compared with the CK group, the Ace index increased 15 days after the application of the two inoculants, and the Ace and Chao 1 indices increased significantly 30 days after application (*p* < 0.05). Compared with the CK group, the Shannon index increased 15 days after the application of the YH-20 microbial inoculant and increased 30 days after the application of the two inoculants. The Simpson index showed no difference among the different treatments and different time periods. In the CK group, the Chao1 index at 30 days was significantly lower than that at 15 days (*p* < 0.05), while the ACE and Shannon indices were slightly lower at 30 days than at 15 days, but the decrease was not significant. In the YH-18 treatment, the ACE, Chao 1, Shannon, and Simpson indices at 30 days after inoculation were not different from those at 15 days after inoculation. In the YH-20 treatment, the ACE, Chao 1, and Simpson indices 30 days after inoculation were not different from those 15 days after inoculation, but the Shannon index 30 days after inoculation was slightly lower than that 15 days after inoculation, although the effect was not significant. The results showed that within 30 days, the diversity and richness of the rhizosphere soil microbial community in the three treatments showed a downward trend, but the inoculation treatment effectively slowed the decline in diversity and richness within 30 days (Table 2).

The results of the correlation analysis between the soil properties and the richness and diversity indices of the bacterial communities are shown in Table 3. The ACE, Chao, and Shannon indices of the bacterial community were positively correlated with OM, AK, and soil pH. The ACE and Chao 1 indices were positively correlated with AN. In addition, the Shannon and Simpson indices were positively correlated with AP.

### 2.5. Bacterial Community Composition at the Taxonomic Level after Inoculation

Except for the unclassified sequences, high-throughput sequences were attributed to each of the 36 bacterial groups, and the top 20 dominant phyla are shown in Figure 3. The dominant phyla whose relative abundances were greater than 1% included Proteobacteria (39.42–46.46%), Bacteroidetes (13.45–23.68%), Acidobacteria (15.35–31.07%), Gemmatimonadetes (3.79–5.12%), Actinobacteria (1.39–1.92%), Patescibacteria (0.95–1.21%), Chloroflexi (1.30–1.84%), Verrucomicrobia (1.13–2.08%), Nitrospirae (0.67–1.46%), Latescibacteria (0.22–1.11%), and Rokubacteria (0.18–1.30%) (Figure 3).

Among these dominant bacteria, compared with the non-inoculated CK, the rhizosphere soil inoculated with YH-18 and YH-20 was significantly enriched in Proteobacteria, whereas Verrucomicrobia, Nitrospirae, and Latescibacteria were significantly reduced 15 days after inoculation. At 30 days, Proteobacteria and Bacteroidetes were significantly enriched, while Acidobacteria, Verrucomicrobia, Latescibacteria, and Rokubacteria were reduced. In the natural succession of the rhizosphere bacterial community of the CK group within 30 days, Bacteroidetes and Nitrospirae decreased significantly, while Acidobacteria, Latescibacteria, and Rokubacteria increased significantly. The application of inoculants slowed the changes in Bacteroidetes, Acidobacteria, Latescibacteria, and Rokubacteria (Figure 3).

To elucidate the peach seedling-soil feedback processes, the differences in the rhizosphere soil microbial populations were compared at the genus level following inoculation. Among the dominant genera, compared with the non-inoculated CK, inoculation with YH-18 and YH-20 significantly enriched *Sphingomonas*, *Pseudomonas*, and *Devosia* and reduced *Bryobacter*, *Chryseolinea*, and *Nitrospira* in the rhizosphere soil on the 15th day. On the 30th day, *Pseudomonas*, *Flavobacterium*, *Arenimonas*, *Aquicella*, *Devosia*, *Novosphingobium*, and *Ferruginibacter* were significantly enriched, while *RB41, Haliangium*, and *MND1* decreased significantly. In the natural succession of the rhizosphere bacterial community of the CK group within 30 days, *Gemmatimonas* and *Haliangium* increased, and *Pseudomonas*, *Flavobacterium*, *Nitrospira*, *Ohtaekwangia*, and *Bacillus* decreased. The use of inoculants slowed the decrease in *Pseudomonas*, *Flavobacterium*, and *Bacillus* (Figure 4).

### 2.6. Community Composition at the OTU Level after Inoculation

A Venn diagram was used to show the similarities and differences in the bacterial community composition between the treatments; fifteen days after inoculation, the three treatments shared 74.17% of the OTUs (3159 OTUs). YH-18 and YH-20 shared the highest number of OTUs (93.76%), whereas YH-18, YH-20 and the non-inoculated CK shared fewer OTUs (91.58% and 92.70%). In addition, the non-inoculated CK had the highest number of unique OTUs (4.44%) (Figure 5A). Thirty days after inoculation, the three treatments shared 68.07% of the OTUs (2884 OTUs). YH-18 and YH-20 shared the highest number of OTUs (76.96%), whereas YH-18, YH-20, and the non-inoculated CK shared fewer OTUs (71.46% and 72.74%). Moreover, YH-18 and YH-20 harbored the highest number of unique OTUs (5.10% and 6.59%) (Figure 5B). In the natural succession of the rhizosphere bacterial community within 30 days, the number of OTUs in the three treatments decreased, but the number of OTUs in the two inoculation groups was significantly higher than that in the CK group on the 15th and 30th days after inoculation (Figure 5A,B).

### 2.7. A PCoA Plot Based on Bray–Curtis Distances Provided a Visualization of the Microbial

The community composition, depending on the inoculant and the sampling period, divides the bacterial community into six clusters based on the three treatments (CK, YH-18, and YH-20) and two time periods (15 and 30 days after inoculation). PERM ANOVA confirmed that at 15 and 30 days, there were significant differences in the soil microbial communities between the two inoculated treatments and the non-inoculated CK, while the differences between the two inoculated treatments were not significant. There were also significant differences in the soil microbial communities within the same treatments at different time periods (*p* < 0.05, Figure 6).

### 2.8. Correlation between Environmental Parameters and Bacterial Community Composition at the Genus Level

The relationship between the soil environmental parameters and the relative abundances of dominant bacteria at the genus level at 30 days was assessed using RDA. The results showed that the relative abundances of *Pseudomonas, Flavobacterium, Arenimonas*, and *Ferruginibacter* were positively correlated with AP, AK, AN, OM, and pH (Figure 7).

## 3. Discussion

The two *B. velezensis* microbial inoculants assessed in this study effectively promoted ground diameter and plant height growth in different time periods within 180 days and simultaneously increased the leaf area and chlorophyll content. There have been many international studies on the growth-promoting mechanism of *B. velezensis* and *B. velezensis* Lle-9, which effectively promoted the growth of lilies by producing organic acids, IAA, and siderophores and by fixing nitrogen and dissolving phosphate [28]. Xue et al. [29] found that the application of *B. velezensis* A3 improved the structure of the soil microbial community, promoted the process of soil nitrification, and then increased crop yield. The functions and growth-promoting effects of the two *B. velezensis* strains in this study are consistent with reports that the strains can promote plant growth by dissolving phosphorus, fixing nitrogen, and regulating soil microecology [30,31].

Siebielec’s research found that the use of bacterial inoculants and soil amendments in soils and wastes contaminated with metals can improve plant growth and increase the availability of nitrate in the soil and that inoculants can play a role in the nitrogen cycle [32]. Shi et al. inoculated *B. pumilus* HR10 into *Carya illinoinensis*, and the levels of available phosphorus and potassium in the rhizosphere soil and the total potassium content in the plant roots were significantly increased [33]. These studies have shown that the application of microbial inoculants can increase organic matter and available nutrient content [34], and the same trend was observed in this study. Thirty and 90 days after inoculation with *B. velezensis*, the soil organic matter, pH, and available nutrients were improved to some extent compared to the non-inoculated control. Microbes can transform insoluble nutrients in soil and fertilizer into forms that can be directly absorbed and utilized by plants through acidolysis, enzymolysis, and polysaccharide complex dissolution [35]. In this study, there was a correlation between the soil environmental parameters and seedling growth. Therefore, the increase in seedling growth may be related to the improvement in soil nutrients. In addition, the acidic soil used in this experiment, which had been under continuous cropping for many years, showed an effective increase in pH after inoculation with *Bacillus* sp., which is consistent with the results of Daraz et al. [36]. We speculate that this may be because the inoculants or altered rhizosphere microbial communities produce more alkaline substances, such as proteins and lipids, during metabolism, so the soil pH rises. This result also provides theoretical support for microbial inoculants to improve continuous cropping soils.

Our results showed that, compared with the control in the same period, the two inoculation treatments improved the α diversity of bacteria in two time periods after inoculation, and the decrease in microbial diversity and abundance in the rhizosphere slowed down within 30 days after inoculation. This shows that the use of inoculants has a significant positive impact on the bacterial community in the peach rhizosphere, which can slow the decline in microbial richness and diversity in the rhizosphere soil. This is consistent with many studies showing that inoculation with *Bacillus* can improve the α diversity of plant rhizosphere microorganisms [15,37]. Correlation analysis showed that there was a significant positive correlation between the soil nutrient content and the diversity of bacteria. The RDA results showed that the relative abundances of *Pseudomonas, Flavobacterium*, and other dominant bacteria genera were positively correlated with the soil nutrient content. The research of Ren et al. also found that soil NH_4_^+^-N was significantly related to the soil community composition of the dominant bacterial genus after the application of biochar and PGPR [38]. This showed that in addition to a large number of microorganisms enriched by inoculation, the release of soil nutrients was enhanced to a certain extent, and the rhizosphere environmental parameters and rhizosphere dominant genus interact closely with each other. As a result of the change in the microbial community structure in the rhizosphere soil of the plants after inoculation, the changed rhizosphere microorganisms acted together to release soil nutrients and promote the nutrient absorption and utilization of plants.

In addition, inoculation with growth-promoting bacteria will lead to the enrichment of some beneficial bacteria in rhizosphere soil [39]. Proteobacteria in soil are usually considered the dominant phylum with the highest genetic and metabolic diversity. The research results of Fierer et al. [40] showed that the relative abundance of symbiotic microorganisms (such as Bacteroidetes) was very high in an environment with high soil organic matter and nutrients. Research conducted by Yu et al. [41] showed that most members of Bacteroidetes mainly decompose cellulose and refractory aromatic compounds, which are very important in soil mineralization. Proteobacteria and Bacteroidetes, as important phyla, were the main participants in the soil nitrogen cycle [42] and were both significantly enriched after inoculation. In the CK group rhizosphere soil, the abundance of Acidobacteria obviously increased within 30 days, while the inoculation treatment controlled the number of Acidobacteria, which was obviously lower 30 days after inoculation than in the control, which may be related to the control of soil acidification by the inoculation treatment and the Acidobacteria being more suited to acidic soil [43].

At the genus level, *Pseudomonas* is widely used for various plants as a plant growth promoter. Sun et al. (2021) proposed that inoculation with *B. velezensis* could recruit beneficial *Pseudomonas* to keep plants healthy, promote plant growth and help plants alleviate salt stress. Moreover, many *Pseudomonas* species can effectively control crown gall [44]. *Flavobacterium* has been shown to promote growth and may be involved in biological phosphorus removal [45]. The *Arenimonas* species can catalyze acid and alkaline phosphatase, esterase (C4), esterase lipase (C8), lipase (C14), and arylamidase [46,47,48]. According to genome sequencing information, *Arenimonas* is capable of metabolizing casein, gelatin, β-hydroxybutyric acid, tyrosine, L-alaninamide, L-glutamic acid, and glycyl-L-glutamic acid [49]. According to RDA, *Pseudomonas*, *Flavobacterium*, and *Arenimonas* showed significant positive correlations with soil nutrients. These strains play an important role in the decomposition of complex organic matter and the transformation of nitrogen, phosphorus, and potassium in the soil. By improving the utilization rate of nutrients, these strains are beneficial to the growth of plants. This result enriches our understanding of the principle of growth-promoting bacteria at this stage and lays a theoretical foundation for the further production of growth-promoting bacteria as high-efficiency soil remediation agents.

In non-sterile soil, it is very common for the number of inoculated bacteria to decrease rapidly. During the plant growth period, reinoculation at regular intervals is necessary to maintain effective bacterial density in the field [50]. In this study, the abundance of *Bacillus* was not high, and it was not the dominant genus among the rhizosphere microorganisms. This may be because the number of other dominant bacteria in the soil was extremely rich [51], and the amount of added *Bacillus* and native *Bacillus* was not enough to exceed that of other native dominant bacteria. The second reason may be the natural death of *Bacillus* after inoculation or flowing away with water, resulting in low abundance. In addition, previous studies have shown that YH-18 and YH-20 can colonize plants, and the study only examined the bacterial population in the soil. It is possible that beneficial bacteria will colonize inside the host plants as endophytes, leading to a decrease in the number of rhizosphere *Bacillus* [52]. Even though the number of rhizosphere *Bacillus* was small and showed a decreasing trend within one month after inoculation, the decreasing rate of the two inoculation treatments was significantly lower than that of the control treatment without inoculation. This may have been due to the mass reproduction of the inoculant within 30 days or the recruitment of bacteria of the same genus after the use of the inoculant, maintaining the abundance of *Bacillus*. In addition, within 30 days, the α diversity of the rhizosphere soil microbial community in the two inoculation treatments was significantly higher than that in the control, and the specific bacterial phyla and genera changed significantly. The results showed that within 30 days of inoculation, the regulatory effect of the inoculants was obvious, and the soil environment changed dramatically. This is consistent with a previous finding that the use of inoculants can cause short-term changes in agricultural soil [53]. However, the long-term impact on the rhizosphere microbial community after inoculation needs to be further explored. However, in this study, inoculation of these two strains only at the time of planting still significantly improved the growth of the peach seedlings within 180 days, which means that there may be a specific plant growth-promoting mechanism independent of the concentration of inoculant in the inoculated strains that can improve the soil environment for a long time, which is consistent with the research results of Chen et al. [54] and Kang et al. [55].

The use of inoculants will adjust the structure of the rhizosphere microbial community, and the interaction of microorganisms can affect the function of the inoculants used. These results showed that, compared with the uninoculated control, the two inoculants better regulated and controlled the bacterial community structure and function of *P. davidiana*. In addition, in all of the results, according to the analyses of richness and diversity of OTUs (α diversity and β diversity), critical phyla, and genus types, the influence of the YH-18 inoculant was very similar to that of the YH-20 inoculant. From the perspective of the bacterial community composition, the enrichment and reduction of different types of soil microorganisms at the phylum and genus levels caused by the two inoculation treatments after 15 days and 30 days were very similar, and the change direction was roughly the same. This is consistent with the enrichment and reduction of microbial species in rhizosphere soil after inoculation with *B. velensis* found in another study [56]. This indicates that the regulatory effects of *B. velensis* inoculants on plant rhizosphere microbial communities may be similar, possibly causing the aggregation and reduction of certain bacteria, but at different intensities. However, the results of this experiment are also different from some studies on rhizosphere changes after inoculation with *B. velensis*, which may be due to differences in microbial flora that can be recruited by different plants [57] and great differences in the original microbial flora of different soil environments [58]. It is also possible that this pot experiment was carried out in an environment similar to that of isolated bacteria, so the change direction of the microbial community was relatively consistent. However, these findings are largely limited by microbial sequencing methods, environmental factors, and the differences among individual plants. Whether *B. velensis* has the same specificity in terms of changing the microbial community structure in rhizosphere soil still needs to be explored and studied in additional experiments.

In this study, the effects of two different inoculation treatments on the growth of *P. davidiana* were analyzed and compared by inoculating the seedlings with *B. velensis* at planting. The results showed that inoculation with the microbial inoculants during replanting promoted the growth and quality of *P. davidiana* and improved the soil environmental problems caused by improper cultivation. Moreover, inoculation with microbial inoculants can effectively improve the microbial community structure of rhizosphere soil in a short time and increase the number of beneficial bacteria in the rhizosphere, thus achieving a long-term growth-promoting effect. This technology has the advantages of low cost and sustainability and is suitable for large-scale use. However, further research is needed to explore the key signals and specific mechanisms by which beneficial microorganisms regulate peach seedling growth.

## 4. Materials and Methods

### 4.1. Overview of the Potted Seedling Test Site

The test site is located in the village of Baiyang in the town of LvXiang, Jinshan District, Shanghai, China (121° E, 30° N, 10 m above sea level). Jinshan District is located south of the Yangtze River. It has a subtropical monsoon climate, with an annual average temperature of 15.8 °C and an annual average precipitation of 1178.2 mm. The soil pH of the test site is 6.0–6.2 (data provided on a trial basis).

### 4.2. Experimental Materials

The potted plants in this experiment were annual *P. davidiana* seedlings. The test fertilizer was a rich organic farm fertilizer provided by Shandong Feiwo Agricultural Materials Co., Ltd., which mainly consists of chicken manure, soybean meal, and hemp cake, with N, P, and K contents higher than 5% and an organic matter content higher than 45%. The potting soil was continuously cropped soil from the Flat Peach Research Institute, where peach trees had been planted for more than 20 years. 

The strains used in the experiment were two plant growth-promoting strains, YH-18 and YH-20, which were isolated from cherry blossom tissues in Shanghai and were identified as *B. velezensis* (previously named *B. methylotrophus* YH-18 and *B. amyloliquefaciens* YH-20) by 16S rRNA analyzes (accession numbers SUB11813965 and MH894222). The two strains have different plant-growth-promoting abilities (Table 4). The strains were stored in the forest pathology laboratory of Nanjing Forestry University. YH-18 and YH-20 were grown on nutrient agar (NA) liquid medium in an artificial vibrating incubator at 28 °C and 200 rpm for 24 h to prepare bacterial inoculants.

### 4.3. Pot Experiment

This experiment was carried out in an open-air greenhouse at the Flat Peach Research Institute in April 2020. The fertilizer was mixed with soil at a ratio of 1:4, and the organic matter, available phosphorus, available potassium, and alkali-hydrolysable nitrogen contents of the mixed soil were 53.07 g/kg, 346.33 mg/kg, 1129 mg/kg, and 377 mg/kg, respectively. The *P. davidiana* seedlings were planted in 37.5-cm diameter × 40-cm high plastic pots, and the plants were planted and cultured under three different treatment conditions for 180 days. For the YH-18 group, the rhizosphere of each *P. davidiana* seedling was inoculated with 100 mL of YH-18 suspension (1.0 × 10^8^ CFU/mL). For the YH-20 group, the rhizosphere of each *P. davidiana* seedling was inoculated with 100 mL of YH-20 suspension (1.0 × 10^8^ CFU/mL). Finally, the control group (CK) contained the test substrate without inoculation. The root irrigation method was used to inoculate *P. davidiana* seedlings. Four replicates were evenly set for each treatment with 24 seedlings. All of the plants were kept well irrigated and protected from weeds. 

### 4.4. Seedling Height and Ground Diameter

Twenty-four seedlings of *P. davidiana* were randomly selected for each repetition to determine the seedling height and ground diameter with a tape measure (0.1 cm accuracy) and a Vernier caliper (0.01 mm accuracy), respectively, when the inoculum was initially applied. Thirty, 90, and 180 days after the application of the inoculum, the height and ground diameter of the peach seedlings were measured, and the net growth within each period was calculated.

### 4.5. Leaf Area and Chlorophyll Content

Before the leaves fell (10 May 2020 and 10 August 2020), 24 seedlings of *P. davidiana* were randomly selected for each repetition to determine the leaf index. One and three months after inoculation, the chlorophyll content and leaf area of the upper, middle, and lower leaves without mechanical damage were measured with a SPAD chlorophyll meter (SPAD-502, Konica, Minolta Sensing, Inc., Sakai, Osaka, Japan) and a leaf area meter (LA 211, Systronics., New Delhi, India), respectively.

### 4.6. Soil Sample Collection

At 0, 15, 30, and 90 days after inoculation, a small shovel was used to expose part of the rhizosphere on the fixed side of each potted plant. A large amount of soil without roots was removed, the soil near the roots was removed with a knife, and the rhizosphere soil (the 0–5 mm soil adhered to the roots) was collected with a sterile brush [59]. During each sampling, a total of 12 seedlings of rhizosphere soil were collected to conduct experiments with four biological replicates of each treatment. The soil samples were stored in sterilized and sealed polyethylene bags. Some samples were freeze-dried at 0, 30, and 90 days to determine the soil properties. In addition, the rhizosphere soils collected at 15 days and 30 days were stored at −80 °C for DNA extraction and high-throughput sequencing analysis.

### 4.7. Determination of Soil Properties

The soil pH was measured by a pH meter (soil-water ratio 2.5:1). After air-drying, the soil hydrolysis nitrogen content was determined by the alkaline hydrolysis-diffusion method [60]. The available phosphorus (AP) content was analyzed by the molybdenum antimony colorimetric method with a UV/visible spectrophotometer (UV-2450/2550, Japan) after ammonium bicarbonate (NaHCO_3_) extraction [61]. The available potassium (AK) content was analyzed by flame photometry (Model 420 Flame Photometer, Sherwood Scientific Ltd., Cambridge, UK) after ammonium acetate (CH_3_COONH_4_) extraction [62]. The soil organic matter content was determined by potassium dichromate oxidation-external heating [63].

### 4.8. Total Bacterial DNA Extraction, Polymerase Chain Reaction (PCR) and 16S rRNA Amplification Based on High-Throughput Sequencing

Fresh rhizosphere soil (0.5 g) was collected, and the total soil DNA was extracted by a Power Soil DNA Extraction Kit (MOBIO Laboratories Inc., Carlsbad, CA, USA). The concentration and purity of the DNA were determined by a Nanodrop 2000 spectrophotometer (Thermo Fisher Scientific., Waltham, MA, USA), and the integrity of the DNA was detected by 1% (*m*/*v*) agarose gel electrophoresis.

Then, the extracted DNA was stored at −20 °C for subsequent analysis. With the same amount of DNA extracted from each sample as the amplification template, 343F (5’-TACGRAGGCAGCAG-3’) and 798R (5’-AGGGTATCTAATCCT-3’) were used to amplify the bacterial 16S rRNA gene for the V3-V4 variable region [64]. The PCR reaction mix (30 μL) contained 2×Gflex PCR Buffer (15 µL), forward primer (5 µM; 1.0 µL), reverse primer (5 µM; 1.0 µL), Tks Gflex DNA Polymerase (0.6 µL), template DNA (50 ng), and ddH_2_O (up to 30 µL). The reaction conditions and amplification procedures were performed as follows: initial denaturation at 94 °C for 5 min; 26 cycles of denaturation at 94 °C for 30 s, annealing at 56 °C for 30 s and extension at 72 °C for 20 s; and a single extension at 72 °C for 5 min, ending at 10 °C. The PCR products were detected by electrophoresis and then purified by Agencourt AM Pure XP beads (Beckman Coulter Co., Brea, CA, USA) and quantified using QuantiFluor™-ST (Promega, Madison, WI, USA) according to the manufacturer’s protocol. The libraries were sequenced with the Illumina MiSeq platform (Illumina Inc., San Diego, CA; GENEWIZ, Inc; Suzhou, China). The raw sequences were submitted to the National Center for Biotechnology Information (NCBI) under accession No. PRJNA 847408.

### 4.9. Bioinformatics Analysis

The raw sequencing data were in FASTQ format. The paired-end reads were preprocessed using Trimmomatic software to detect and remove ambiguous bases (N) [65]. Forward and reverse reads of the same sequence were merged using FLASH v1.2.5 such that there was more than 200 bp of overlap and <0.25 mismatches [66]. Chimeric sequences were detected and removed with the UCHIME algorithm [67]. The sequences were clustered into operational taxonomic units (OTUs) with a 97% identity cutoff using VSEARCH 2.4.2, and their taxonomic affiliation was assigned using the RDP 16S rRNA reference database [68]. The alpha diversity was analyzed using QIIME, which included calculations of the ACE, Chao 1, Shannon, and Simpson indices [69]. Similarly, the beta diversity was estimated by computing the unweighted UniFrac distance and was visualized via principal coordinate analysis (PCoA) [70]. The typical principal coordinate method (CAP) and PERMANOVA method were used to visually detect the significant differences in community structure and functional structure [71].

### 4.10. Statistical Analysis

The experimental data were analyzed using SPSS 22.0 (SPSS Inc., Chicago, IL. USA), and the significant differences were analyzed by ANOVA and Duncan’s test (*p* < 0.05). Complete data visualization was performed with Origin Pro 8.5 software (Northampton, MA 01060 USA). Correlation analyses were calculated using Pearson correlation analysis SPSS software, version 22 (SPSS Inc., Chicago, IL. USA).

## Figures and Tables

**Figure 1 ijms-23-13639-f001:**
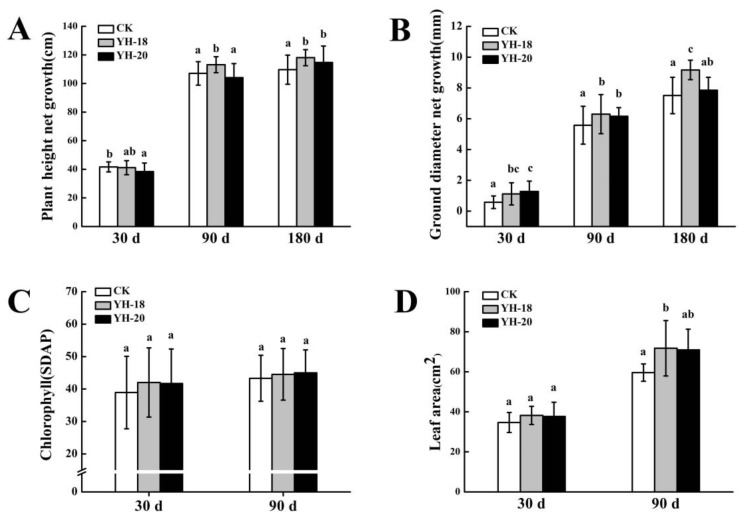
Effects of inoculation with YH-18 and YH-20 on (**A**) ground diameter growth, (**B**) height growth, (**C**) chlorophyll content and (**D**) leaf area of *P. davidiana*. Different letters indicate significant differences between different treatments in a group within the same period at a confidence level of *p* < 0.05.

**Figure 2 ijms-23-13639-f002:**
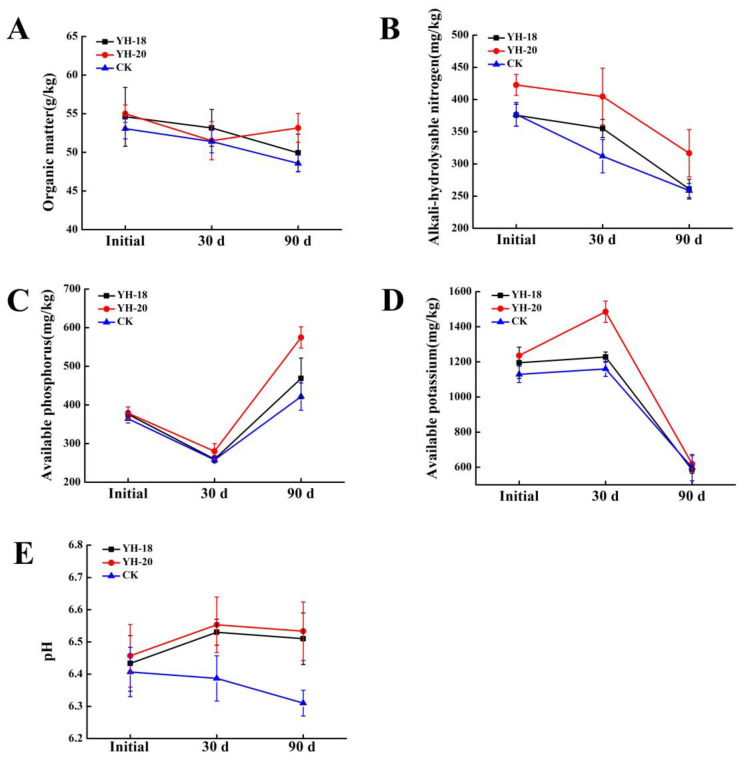
Effects of YH-18 and YH-20 microbial inoculants on (**A**) OM, (**B**) AN, (**C**) AP, (**D**) AK, and (**E**) pH in soil.

**Figure 3 ijms-23-13639-f003:**
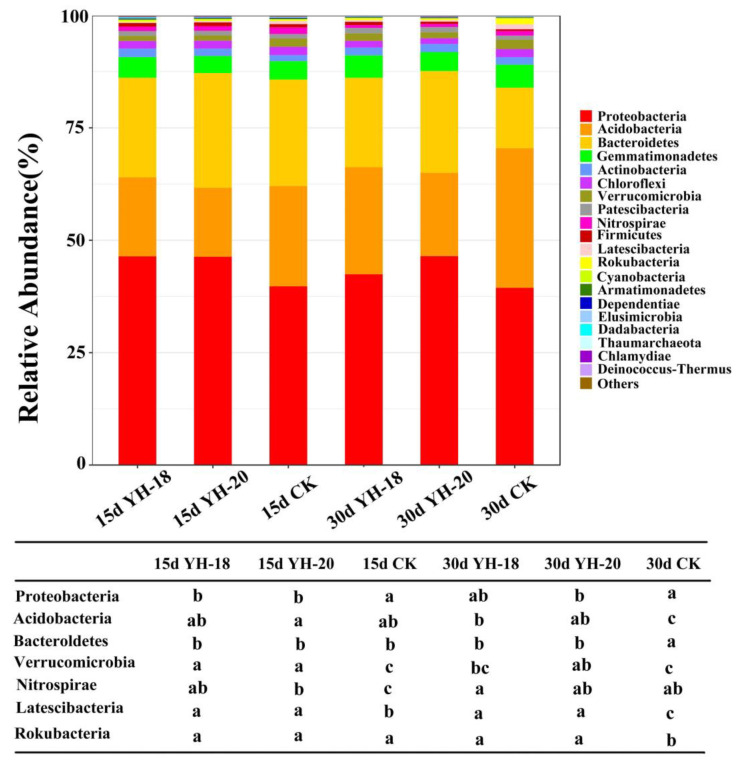
The 20 most abundant bacterial groups at the phylum level.

**Figure 4 ijms-23-13639-f004:**
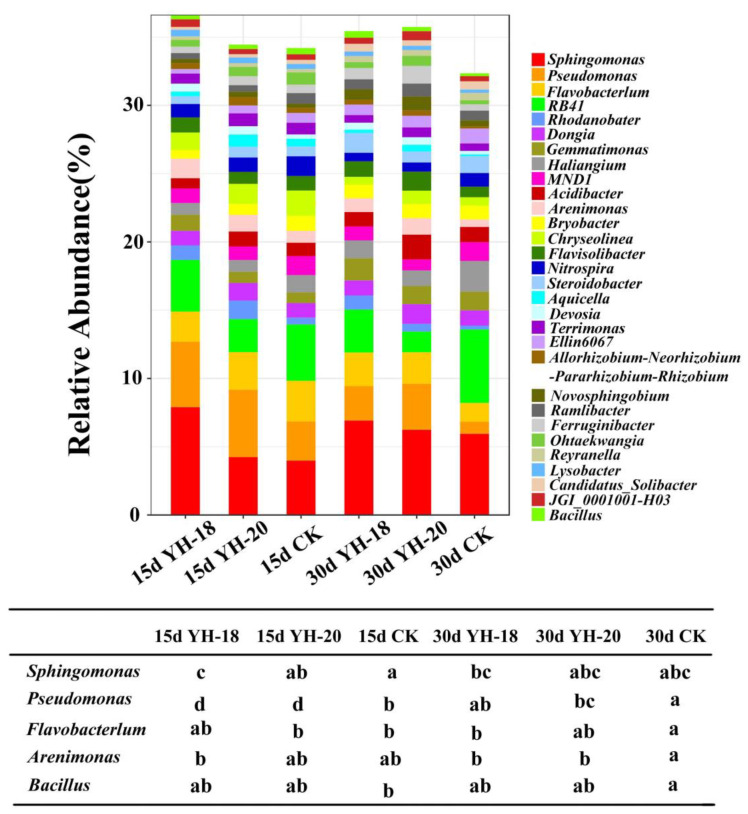
The 30 most abundant bacterial groups at the genus level.

**Figure 5 ijms-23-13639-f005:**
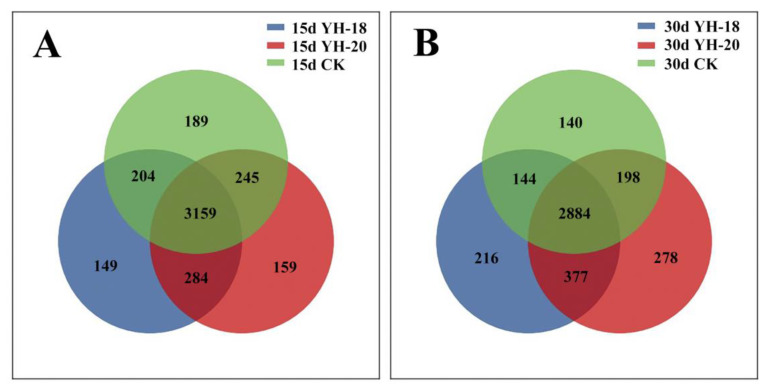
Venn diagram of different combinations: (**A**) Venn diagram of different treatments 15 days after inoculation. (**B**) Venn diagram of different treatments 30 days after inoculation.

**Figure 6 ijms-23-13639-f006:**
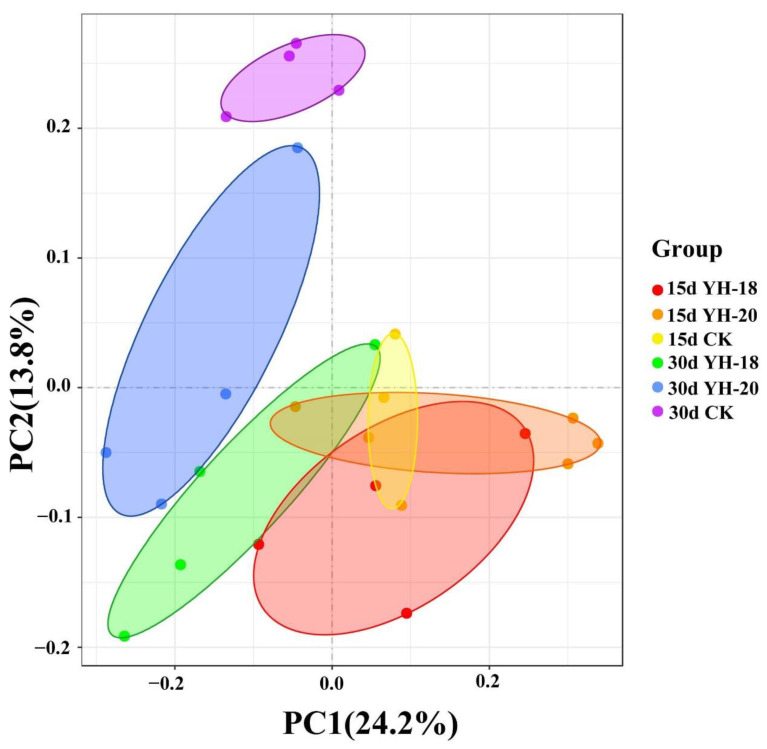
Principal coordinate analysis of 16S rRNA genes of total bacteria based on the weighted similarity index at 97% identity (operational taxonomic unit level). PC1 and PC2 explained 25.9% and 12.3%, respectively, of the variance.

**Figure 7 ijms-23-13639-f007:**
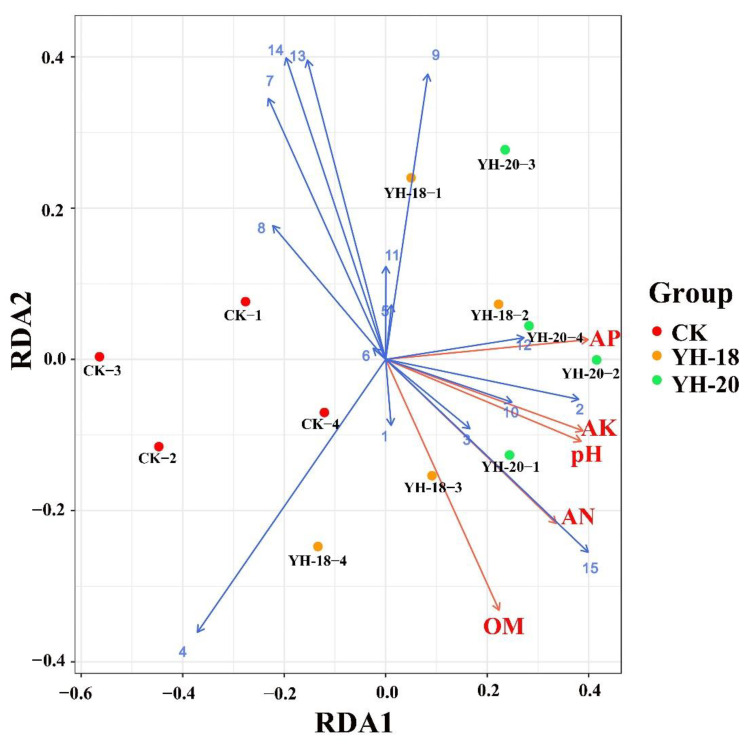
Redundancy analysis of the soil bacterial community composition at the genus level and soil environmental parameters. Soil environmental parameters are represented by red lines and bacterial genera are represented by blue lines (1: *Sphingomonas*, 2: *Pseudomonas*, 3: *Flavobacterium*, 4: *RB41*, 5: *Dongia*, 6: *Gemmatimonas*, 7: *Haliangium*, 8: *MND1*, 9: *Acidibacter*, 10: *Arenimonas*, 11: *Bryobacter*, 12: *Flavisolibacter*, 13: *Ellin6067*; 14: *Steroidobacter*, 15: *Ferruginibacter*).

**Table 1 ijms-23-13639-t001:** Pearson’s analysis between seedling growth correlation indicators and soil nutrient content.

	Ground Net Diameter Growth	Plant Height Net Growth	Leaf Area	Chlorophyll Content
OM	0.198	0.718 *	0.471	0.211
AN	0.629	−0.224	0.547	0.476
AK	0.747 *	−0.244	0.614	0.627
AP	0.421	0.356	0.531	0.38
pH	0.832 **	0.028	0.861 **	0.785 *

* Significant correlation at the 0.05 level. ** Significant correlation at the 0.01 level.

**Table 2 ijms-23-13639-t002:** Richness and diversity indices of bacteria in different treatments.

Treatment	Ace	Chao1	Shannon	Simpson	Coverage
15-YH-18	3278.88 ± 52.11 b	3327.92 ± 70.43 b	9.65 ± 0.07 ab	0.9963 ± 0.001 a	0.9753 ± 0.0005 a
15-YH-20	3325.81 ± 149.77 b	3362.47 ± 150.70 b	9.75 ± 0.16 b	0.9967 ± 0.001 a	0.975 ± 0.0008 a
15-CK	3290.15 ± 82.13 ab	3307.20 ± 75.65 b	9.66 ± 0.08 ab	0.9967 ± 0.001 a	0.975 ± 0.0008 a
30-YH-18	3085.22 ± 161.97 b	3125.67 ± 157.64 b	9.53 ± 0.16 ab	0.9963 ± 0.001 a	0.977 ± 0.0008 bc
30-YH-20	3187.30 ± 281.74 b	3189.75 ± 285.43 b	9.60 ± 0.34 ab	0.9968 ± 0.001 a	0.9757 ± 0.0017 ab
30-CK	2861.84 ± 132.61 a	2867.85 ± 134.50 a	9.38 ± 0.16 a	0.996 ± 0.001 a	0.9785 ± 0.0013 c

Values within each column followed by different lowercase letters are significant at *p* < 0.05.

**Table 3 ijms-23-13639-t003:** Correlations between soil properties and microbial richness and diversity indices.

Treatment	Ace	Chao1	Shannon	Simpson
OM	0.595 *	0.590 *	0.668 *	0.328
AN	0.692 *	0.658 *	0.545	0.219
AK	0.681 *	0.635 *	0.615 *	0.514
AP	0.532	0.504	0.633 *	0.637 *
pH	0.679 *	0.679 *	0.594 *	0.362

* significant at the 0.05 level (*p* < 0.05).

**Table 4 ijms-23-13639-t004:** Growth promoting characteristics of YH-18 and YH-20.

Treatment	Salt Tolerance	Alkali Tolerance	Phosphate Solubilization Capacity (μg/mL)	Potassium Solubilization Capacity (μg/mL)	IAA Production (μg/mL)	Siderophore Production	Nitrogenase Activity
YH-18	9%	10	143.07	14.63	33.59	+	+
YH-20	9%	9	157.75	12.50	17.40	+	+

“+” means positive.

## Data Availability

The raw data supporting the conclusions of this article will be made available by the authors upon request.

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
