# Peer review of "Effects of Two *Bacillus Velezensis* Microbial Inoculants on the Growth and Rhizosphere Soil Environment of *Prunus davidiana"

_ijms, 2022, doi:10.3390/ijms232113639_

Round 1

Reviewer 1 Report

The paper focuses on a very relevant issue which is the use of microbial inoculants, as harmless, efficient and environmentally friendly alternatives to synthetic products in agricultural production. This type of knowledge is worth to be promoted since it helps to understand and prove the effects of inoculants.

The study design is rather simple but this is not a weakness of the paper since it helps the clarity of the results obtained using advanced biodiversity determinations.

The authors provided interesting results that showed positive impact of inoculation with Bacillus strains on the growth of Prunus even if this was relatively short term study for tree plan species. The inoculation also altered soil microbiome which is well presented using the biodiversity data.  The application of genetic studies made it possible to know the composition of the population in the experiment both at the phylum and the genus level.

The article presents in a legible and transparent manner the material and methods used in the research. The methodology is clear and described concisely. Introduction section is comprehensive and is written in an concise and clear manner. The literature is well-chosen and the conclusions clearly refer to the conducted research.

Stimulation of nutrient availability by the inoculation is an interesting observation. Especially stimulation of potassium is rarely reported. Please consider more references on the effect of inoculation n nutrient availability, for example this reference dealing with contaminated soils https://www.mdpi.com/2073-4395/11/10/2064.

There are, however some points for improvement, please consider them. Therefore I recommend minor revision of the paper.

Minor issues to be corrected:

·       Conclusions

Due to the lack of clearly specified conclusions in this study, we do not fully know what message is to be sent by the authors.

I would propose at least two main conclusions at the end of the discussion section based on the research results, which will show the reader the achieved result.

·       Figure 7

2.8. Correlation between environmental parameters and the composition of bacterial communities at the genus level (line 224) - is there any literature data on the described correlation? If so, please provide some citations? please extend line 264.

·       Material and method

4.9. Bioinformatics analysis (line 437) please provide literature citations to the indicators used

·       Material and method

4.7. Determination of soil properties - please provide the apparatus on which the measurements were made available phosphorus and potassium and soil organic matter

Please also try to discuss the following interesting observations. Currently the Authors only mention what was observed but do not try to explain why:

-        Increase in soil pH after inoculation compared to the control

-        Lower abundance of Bacillus genus in soils inoculated with Bacillus strains.

This topic is important due to the use of biological preparations based on bacteria in agricultural production is not yet common, but it may turn out to be necessary facing the need for more sustainable practices. In summary, the paper is worth publishing in the Journal.

Author Response

Response to Reviewer 1 Comments

Point 1. Stimulation of nutrient availability by the inoculation is an interesting observation. Especially stimulation of potassium is rarely reported. Please consider more references on the effect of inoculation N nutrient availability, for example this reference dealing with contaminated soils https://www.mdpi.com/2073-4395/11/10/2064.

Response 1: Thank you for your valuable suggestion. We have added more references about the impact of inoculation on K and N nutrient supply to line 252-257.

Siebielec's research found that the use of bacterial inoculants and soil amendments in soils and wastes contaminated with metals can improve plant growth and increase the availability of nitrate in the soil, and that inoculants can play a role in the nitrogen cycle [32]. Shi et al. inoculated B. pumilus HR10 into Carya illinoinensis, and the levels of available phosphorus and potassium in the rhizosphere soil and the total potassium content in the plant roots were significantly increased [33]. (line252-257)

Point 2. Due to the lack of clearly specified conclusions in this study, we do not fully know what message is to be sent by the authors.

I would propose at least two main conclusions at the end of the discussion section based on the research results, which will show the reader the achieved result.

Response 2: We appreciate this constructive comment. According to the comments of the reviewers, we have added the main conclusions of relevant research to the end of the discussion section (line 382-392).

In this study, the effects of two different inoculation treatments on the growth of P. davidiana were analyzed and compared by inoculating the seedlings with B. velensis at planting. The results showed that inoculation with the microbial inoculants during replanting promoted the growth and quality of P. davidiana, and improved the soil environmental problems caused by improper cultivation. Moreover, inoculation with microbial inoculants can effectively improve the microbial community structure of rhizosphere soil in a short time and increase the number of beneficial bacteria in the rhizosphere, thus achieving a long-term growth-promoting effect. This technology has the advantages of low cost and sustainability and is suitable for large-scale use. However, further research is needed to explore the key signals and specific mechanisms by which beneficial microorganisms regulate peach seedling growth. (line382-392)

Point 3. Correlation between environmental parameters and the composition of bacterial communities at the genus level (line 224) - is there any literature data on the described correlation? If so, please provide some citations? please extend line 264.

Response 3: Thank you for your important question. We have supplemented the relevant literature reference data to line 282-289 of the manuscript.

RDA results showed that the relative abundances of Pseudomonas, Flavobacterium and other dominant bacteria genus were positively correlated the soil nutrient content. The research of Ren et al. also found that soil NH4+-N was significantly related to the soil community composition of dominant bacterial genus after application of biochar and PGPR [38]. This showed that in addition to a large number of microorganisms enriched by inoculation, the release of soil nutrients was enhanced to a certain extent, and the rhizosphere environmental parameters and rhizosphere dominant genus interact closely with each other. (line282-289)

Point 4. Bioinformatics analysis (line 437) please provide literature citations to the indicators used

Response 4: Thank you for your important reminder. We have added relevant literature citations to the indicators used for bioinformatics analysis in the manuscript (line 498-503).

The alpha diversity was analyzed using QIIME, which included calculations of the ACE, Chao 1, Shannon, and Simpson indices [69]. Similarly, the beta diversity was estimated by computing the unweighted UniFrac distance and was visualized via principal coordinate analysis (PCoA) [70]. The typical principal coordinate method (CAP) and PERMANOVA method were used to visually detect the significant differences in community structure and functional structure [71]. (line 498-503)

Point 5. Determination of soil properties - please provide the apparatus on which the measurements were made available phosphorus and potassium and soil organic matter

Response 5: Thank you for your careful comments. We have added the instruments for measuring phosphorus and potassium content (line 461-467). However, the soil organic matter content is heated by potassium dichromate, and no special measuring tools are used.

The available phosphorus (AP) content was analyzed by the molybdenum antimony colorimetric method with a UV/visible spectrophotometer (UV-2450/2550, Japan) after ammonium bicarbonate (NaHCO3) extraction [61]. The available potassium (AK) content was analyzed by flame photometry (Model 420 Flame Photometer, Sherwood Scientific Ltd, Cambridge, UK) after ammonium acetate (CH3COONH4) extraction [62]. The soil organic matter content was determined by potassium dichromate oxidation-external heating [63]. (line 461-467)

Point 6. Please also try to discuss the following interesting observations. Currently the Authors only mention what was observed but do not try to explain why:

Increase in soil pH after inoculation compared to the control

Response 6: Thank you for your question. We have added the relevant reasons for the increase in the soil pH value after inoculation to the Discussion section of the manuscript in line 266-272.

In addition, the acidic soil used in this experiment, which had been under continuous cropping for many years, showed an effective increase in pH after inoculation with Bacillus sp, which is consistent with the results of Daraz et al [36]. We speculate that this may be because the inoculants or altered rhizosphere microbial communities produce more alkaline substances, such as proteins and lipids, during metabolism, so that the soil pH rises. This result also provides theoretical support for microbial inoculants to improve continuous cropping soils. (line 266-272)

Point 7. Lower abundance of Bacillus genus in soils inoculated with Bacillus strains.

Response 7: Thank you for your question. We have discussed the reasons for the low abundance of Bacillus in rhizosphere microorganisms in line 328-337.

In this study, the abundance of Bacillus was not high, it was not the dominant genus among the rhizosphere microorganisms. This may be because the number of other dominant bacteria in the soil was extremely rich [51], and the amount of added Bacillus and native Bacillus was not enough to exceed that of other native dominant bacteria. The second reason may be the natural death of Bacillus after inoculation or flowing away with water, resulting in low abundance. In addition, previous studies have shown that YH-18 and YH-20 can colonize plants, and the study only examined the bacterial population in soil. It is possible that beneficial bacteria will colonize inside the host plants as endophytes, leading to a decrease in the number of rhizosphere Bacillus [52]. (line 328-337)

Reviewer 2 Report

This manuscript reports on the inoculation of two plant growth promoting Bacillus velezensis strains on growth of Prunus davidiana and bacterial community in the rhizosphere. The content of the manuscript fits well with the aim and scope of the journal and should attract a wide readership. The manuscript is well written and clear representation. Only few comments listed below. Therefore, I recommend a minor revision of this manuscript.

Major points

Line 301-303 “In non-sterile soil, it is very common for the number of inoculated bacteria to decrease rapidly” Since the study only examine bacterial population in soil, it is possible that beneficial bacteria will colonize inside the host plants as endophyte. Please discuss this point.

Line 304-305 The authors claimed that “in this study, the abundance of Bacillus in the non-inoculate treatment gradually decreased within 30 days, while the abundance of Bacillus in the two inoculation treatments remained at a certain level.” From Fig. 4, I saw a decreased for both Bacillus at 30 days. Please clarify this point. What is the possible reason for such decrease?

Please add figure showing growth of Prunus davidiana with and without bacterial inoculation

Please add information on how to prepare bacterial inoculum eg. culture medium, cultivation condition

Line 390 Before the leaves fell,…. Please specify the sampling date

What is the location of the inoculated bacteria in Prunus davidiana?

Please give detail on correlation analysis eg. which program/statistics is used

Please give detail of Pearson alaysis

Minor points

Line 22 Pseudomonas must be italicized

Line 52 Please explain “unreasonable planting”

Line 67 Change “preparation of bacterial preparations” to preparation of bacterial bioinoculants

Line 250 “…with B. velezensis for,…” for what?

Line 277-278 what the authors mean by “Bacteridota”? In the result only Bacterioidetes mentioned

Line 291 Please explain esterase lipase. What is the different between esterase lipase to esterase or lipase

Author Response

Response to Reviewer 2 Comments

Point 1. Line 301-303 “In non-sterile soil, it is very common for the number of inoculated bacteria to decrease rapidly” Since the study only examine bacterial population in soil, it is possible that beneficial bacteria will colonize inside the host plants as endophyte. Please discuss this point.

Response 1: Thank you for your important question. We have supplemented the relevant discussion in line 333-337 in the manuscript.

In addition, previous studies have shown that YH-18 and YH-20 can colonize plants, and the study only examined the bacterial population in soil. It is possible that beneficial bacteria will colonize inside the host plants as endophytes, leading to a decrease in the number of rhizosphere Bacillus [52]. (line 333-337)

Point 2. Line 304-305 The authors claimed that “in this study, the abundance of Bacillus in the non-inoculate treatment gradually decreased within 30 days, while the abundance of Bacillus in the two inoculation treatments remained at a certain level.” From Fig. 4, I saw a decreased for both Bacillus at 30 days. Please clarify this point. What is the possible reason for such a decrease?

Response 2: We apologize for this inadvertently misleading sentence. We corrected the phrasing of the sentence to clarify the exact meaning. The specific corrections are in line 328-340 of the manuscript.

In this study, the abundance of Bacillus was not high, it was not the dominant genus among the rhizosphere microorganisms. This may be because the number of other dominant bacteria in the soil was extremely rich [51], and the amount of added Bacillus and native Bacillus was not enough to exceed that of other native dominant bacteria. The second reason may be the natural death of Bacillus after inoculation or flowing away with water, resulting in low abundance. In addition, previous studies have shown that YH-18 and YH-20 can colonize plants, and the study only examined the bacterial population in soil. It is possible that beneficial bacteria will colonize inside the host plants as endophytes, leading to a decrease in the number of rhizosphere Bacillus [52]. Even though the number of rhizosphere Bacillus was small and showed a decreasing trend within one month after inoculation, the decreasing rate of the two inoculation treatments was significantly lower than that of the control treatment without inoculation. (line 328-340)

Point 3. Please add figure showing growth of Prunus davidiana with and without bacterial inoculation

Response 3: Thank you for your suggestion. However, due to our negligence, the growth of the peach trees was not accurately photographed at 3 months after inoculation; only the whole experimental site was photographed (Supplementary Figure 1). Therefore, it is impossible to clearly see the difference in peach tree growth between the treatment groups with or without inoculation of bacteria due to the unavailability of photos. This may not meet the relevant suggestions put forward by reviewers. However, there has been much research support for inoculating YH-18 and YH-20 to promote seedling growth. We will supplement the relevant literature as follows (Supplementary Figure 2, Supplementary Table 1-3) [1-2]. This shows that the growth-promoting results of our experiment are true and reliable, and we hope that reviewers will consider them appropriate.

Supplementary Figure 1 Effects of the inoculation with YH-18 and YH-20 on Prunus davidiana seedlings grown after 3 months (A) CK group; (B) the YH-18 group; and (C) the YH-20 group.

Supplementary Figure 2 Effects of different inoculation treatments on the biomass of 2-year-old C. illinoinensis seedlings. Effects of the plant growth-promoting rhizobacteria (PGPR) on C. illinoinensis cultivar Pawnee seedlings grown after 60 days (a), the dry weight (b), the growth rate of plant height (c), and ground diameter (d). Different lowercase letters indicate that there are significant differences between inoculation treatments (p < 0. 05)

[1]Shi, J.W.; Lu, L.X.; Shi, H.M.; Ye, J.R. Effects of plant growth-promoting rhizobacteria on the growth and soil microbial community of Carya illinoinensis. Curr Microbiol, 2022, 79, 352. doi: 10.1007/s00284-022-03027-9

Supplementary table 1 The seedling height and diameter of metasequoia in each treatment group

YH-18

YH-20

CK

Plant height net growth (cm)

27.6

35.9

20.3

Height comparative growth rate compare with CK

36.0%

76.8%

/

Plant diameter net growth (mm)

8.9

9.1

6.8

Diameter comparative growth rate compare with CK

30.8%

33.8%

/

Supplementary table 2 The seedling height of Cinnamomum camphora in each treatment group

YH-18

YH-20

CK

Plant height net growth (cm)

32.4

29.7

12.9

Height comparative growth rate compare with CK

151.2%

130.2%

/

Supplementary table 3 The seedling diameter of PopulusL in each treatment group

YH-18

YH-20

CK

Plant diameter net growth (mm)

4.73

5.83

4.66

Diameter comparative growth rate compare with CK

1.5%

25.1%

/

[2]Wei, Q. Application and research of several excellent endophytic bacteria in Shanghai ecological forest. [master’s thesis]. [China]: Nanjing Forestry University. 2017.

Point 4. Please add information on how to prepare bacterial inoculum eg. culture medium, cultivation condition

Response 4: I apologize that the inaccurate writing in the article caused this information to lack clarity. We have revised the statement regarding the preparation of the bacterial inoculum, and the revised information is in line 413-415 of the manuscript.

YH-18 and YH-20 were grown on nutrient agar (NA) liquid medium in an artificial vibrating incubator at 28 °C and 200 rpm for 24 hours to prepare bacterial inoculants. (line 413-415)

Point 5. Line 390 Before the leaves fell, Please specify the sampling date

Response 5: We are extremely grateful to the reviewer for pointing out this issue. We have added the specific sampling date to line 441-442 of the manuscript.

Before the leaves fell (May 10, 2020 and August 10, 2020), 24 seedlings of P. davidiana were randomly selected for each repetition to determine the leaf index. (line 441-442)

Point 6. What is the location of the inoculated bacteria in Prunus davidiana?

Response 6: Apologies for being unclear in these parts of our paper. We have supplemented the inoculation location and inoculation method (line 426-430).

For the YH-18 group, the rhizosphere of each P. davidiana seedling was inoculated with 100 mL of YH-18 suspension (1.0×108 CFU/mL). For the YH-20 group, the rhizosphere of each P. davidiana seedling was inoculated with 100 mL of YH-20 suspension (1.0×108 CFU/mL). Finally, the control group (CK) contained the test substrate without inoculation. The root irrigation method was used to inoculate P. davidiana seedlings. (line 426-430)

Point 7. Please give detail on correlation analysis eg. which program/statistics is used

Response 7: We have added the details of the correlation analysis to line 508-509 of the manuscript.

Correlation analyses were calculated using Pearson correlation analysis SPSS software, version 22 (SPSS Inc., Chicago, IL. USA). (line 508-509)

Point 8. Please give detail of Pearson analysis.

Response 8: We have added the details of Pearson's analysis to Line 508-509 of the manuscript.

Correlation analyses were calculated using Pearson correlation analysis SPSS software, version 22 (SPSS Inc., Chicago, IL. USA). (line 508-509)

Point 9. Line 22 Pseudomonas must be italicized

Response 9: Thank you for your important reminder. We have changed the Latin names of Pseudomonas in the manuscript to italics and confirmed the format of all Latin names in the manuscript (line 22-23).

At the genus level, Sphingomonas and Pseudomonas were significantly enriched at 15 and 30 days, respectively. (line 22-23)

Point 10. Line 52 Please explain “unreasonable planting”

Response 10: We apologize for the confusion in this part of our paper. We have modified the word "irrational planting" in the second paragraph to “irrational cultivation” (line 49-52). In the first paragraph of our manuscript, we described irrational planting, which means intensive cultivation, years of continuous cropping and excessive use of chemical fertilizers.

Many studies have shown that microbial inoculants can induce changes in the plant rhizosphere microbial community and improve soil fertility, thus improving the soil environment and crop growth and reducing the pollution caused by unreasonable cultivation [11-12]. (line 49-52)

Point 11. Line 67 Change “preparation of bacterial preparations” to preparation of bacterial bioinoculants

Response 11: Thank you for your important comment. We have changed this description (line 67).

And B. velezensis can form stable endospores that help them survive in the preparation of bacterial bioinoculants [23]. (line 67).

Point 12. Line 250 “…with B. velezensis for,…” for what?

Response 12: Thank you for your important reminder. We have deleted it and checked the typing of the full text (line 259-261).

Thirty and 90 days after inoculation with B. velezensis, the soil organic matter, pH and available nutrients were improved to some extent compared to the non-inoculated control.(line 259-261)

Point 13. Line 277-278 what the authors mean by “Bacteridota”? In the result only Bacterioidetes mentioned

Response 13: We are extremely grateful to the reviewer for pointing this out. Due to our negligence, there was a typing error. We have corrected this information and checked the typing of the full text (line 296-303).

The research results of Fierer et al. [40] showed that the relative abundance of symbiotic microorganisms (such as Bacteroidetes) was very high in an environment with high soil organic matter and nutrients. Research conducted by Yu et al. [41] showed that most members of Bacteroidetes mainly decompose cellulose and refractory aromatic compounds, which are very important in soil mineralization. Proteobacteria and Bacteroidetes, as important phyla, were the main participants in the soil nitrogen cycle [42] and were both significantly enriched after inoculation. (line 296-303)

Point 14. Line 291 Please explain esterase lipase. What is the different between esterase lipase to esterase or lipase.

Response 14: Thank you for your important question. We have revised the sentence so that readers can understand them better (line 314-315). “esterase lipase (C8)” can also be written as “esterase/lipase (C8)”. Esterase lipase is a kind of esterase. The difference among esterase (C4), esterase lipase (C8) and lipase (C14) lies in the length of the carbon chain of the substrate during hydrolysis. Esterase(C4) is an enzyme that can hydrolyze short-chain fatty acids, esterase lipase (C8) is an enzyme that can hydrolyze medium-chain fatty acids, and lipase (C14) is an enzyme that can hydrolyze long-chain fatty acids.

The Arenimonas species can catalyze acid and alkaline phosphatase, esterase (C4), esterase lipase (C8), lipase (C14), and arylamidase [46-48]. (line 314-315)
